# Inflammation Induces Changes in the Functional Expression of P-gp, BCRP, and MRP2: An Overview of Different Models and Consequences for Drug Disposition

**DOI:** 10.3390/pharmaceutics13101544

**Published:** 2021-09-23

**Authors:** Sonia Saib, Xavier Delavenne

**Affiliations:** 1INSERM U1059, Dysfonction Vasculaire et de l’Hémostase, 42270 Saint-Priest-En-Jarez, France; xavier.delavenne@univ-st-etienne.fr; 2Faculté de Médecine, Université Jean Monnet, 42023 Saint-Etienne, France; 3Laboratoire de Pharmacologie Toxicologie Gaz du Sang, CHU de Saint-Etienne, 42000 Saint-Etienne, France

**Keywords:** inflammation, ABC transporters, pharmacokinetic variability, drug transport

## Abstract

The ATP-binding cassette (ABC) transporters play a key role in drug pharmacokinetics. These membrane transporters expressed within physiological barriers can be a source of pharmacokinetic variability. Changes in ABC transporter expression and functionality may consequently affect the disposition of substrate drugs, resulting in different drug exposure. Inflammation, present in several acute and chronic diseases, has been identified as a source of modulation in drug transporter expression leading to variability in drug response. Its regulation may be particularly dangerous for drugs with a narrow therapeutic index. In this context, numerous in vitro and in vivo models have shown up- or downregulation in the expression and functionality of ABC transporters under inflammatory conditions. Nevertheless, the existence of contradictory data and the lack of standardization for the models used have led to a less conclusive interpretation of these data.

## 1. Introduction

The ATP-binding cassette (ABC) transporter family has been widely investigated in several physiological barriers including the blood-brain barrier (BBB), intestine, liver, and kidney [1,2,3,4]. It is known that the pharmacokinetic properties of many drugs and metabolites can be affected by these transporters. They can be a source of significant drug–drug interactions. This phenomenon results in an inhibition or induction of drug transport, contributing to a change in disposition. Transporter-related pharmacokinetic variability may also be due to genetic polymorphisms. Some genetic variations of ABC transporters were reported to be associated with an alteration of their expression and transport function, involving clinically relevant pharmacokinetics changes [5,6].

Interestingly, ABC transporters can also be modulated by other stimuli. Under certain circumstances such as pathological processes or inflammation, the expression and functionality of drugs transporters can be impacted and modified. During inflammation, epithelial and endothelial cells are exposed to several factors such as the proinflammatory cytokines that may be responsible for alterations in protein synthesis [7,8,9]. The complex signaling cascades induced by these inflammatory mediators have been shown to alter the expression of many proteins, including drug transporters [10,11]. Up- and downregulations have been reported in both preclinical and clinical studies [12,13,14]. However, an overview of regulation pathways and the impact of these modulations on drug disposition remains incomplete.

Faced with the complexity of inflammatory mechanisms, several models of inflammation have been used in order to assess its impact on drug transporters. Among these models, the induction of an inflammatory response in rodents is the most used approach. In addition, numerous other approaches are also found in literature including in vitro and in vivo human models, which have generated a lot of data. Nevertheless, the existence of contradictory results and a lack of standardization across models require a synthesis of data and a comparison between the different models in order to better identify the consequences of inflammation.

The aim of this work was to summarize and discuss available data concerning ABC transporter expression and functionality during inflammation within the different physiological barriers of the organism. Among the 48 functional ABC transporters present in the human genome, this review focused on the most pharmacologically relevant transporters including P-gp (gene *ABCB1*), BCRP (gene *ABCG2*) and MRP2 (gene *ABCC2*). 

## 2. Inflammation Mechanisms

### 2.1. The Acute Phase Response

Inflammation is a defensive mechanism in response to infection, injury, or tissue damage, leading to many systemic and metabolic changes. The first reaction, called acute phase response (APR), is associated with fever, vasodilation, and leukocytosis. This phenomenon lasts 24–48 h and causes the release of proinflammatory cytokines by the immune cells present in tissues (macrophages, dendritic cells, and mast cells) [15,16]. During this stage, cytokines play a major role by coordinating the mechanisms that lead to an inflammatory response. These small soluble peptides (8–50 kDa) have a role in cellular communication between immune cells, but they also influence the immune response according to the detected signal. These cytokines including tumor necrosis factor alpha (TNF-α), interleukin-1 (IL-1), interleukin-6 (IL-6), and interferon (IFN), stimulate the synthesis of other cytokines and chemokines which are involved in the migration of immune cells (leukocytes) to the site of inflammation. Many other mediators participate in the APR in response to the cytokine release, such as lipid mediators (prostaglandins and leukotrienes), histamine and acute phase proteins including C-reactive protein (CRP). The hepatic synthesis of several proteins such as α1-acid glycoprotein and CRP is upregulated, resulting in increased plasma concentrations. This mechanism leads to vascular modifications, diapedesis of leukocytes and phagocytosis. On the other hand, the expression of albumin, transferrin, and cytochrome P450 proteins is decreased [17].

### 2.2. Chronic Inflammation: A Brief Summary

In most cases, the end of an acute inflammation process corresponds to the restoration of damaged tissues. However, in some cases it can turn into chronic inflammatory disease. This chronic response contributes to the severity of several diseases, including chronic renal failure, various autoimmune disorders such as rheumatoid arthritis, inflammatory bowel diseases—in particular Crohn’s disease, but also in metabolic disorders (type 1 and 2 diabetes mellitus) or tumoral diseases. Chronic inflammation is also involved in the severity of persistent bacterial or viral infection (e.g., HIV or hepatitis B). The chronicity is due to the loss of regulatory mechanisms and to the activation and accumulation of macrophages that continually release proinflammatory cytokines. This immune imbalance induces an abnormal exposure of cells to the proinflammatory cytokines and affects their functions. Whether in acute or chronic inflammation, cytokines have a central role in the modulation of gene expression. The release of cytokines can exert systemic effects by reaching the bloodstream and interacting with membrane receptors on epithelial or endothelial cells. This cytokine–receptor interaction induces a complex signaling cascade, leading to a transduction of signal to the nucleus [18,19,20]. In response to the signal, many nuclear receptors (NRs) are involved during inflammation, such as the pregnane X receptor (PXR) or the constitutive androstane receptor (CAR), which act as transcription factors [21,22,23,24]. These complex regulation pathways lead to the regulation of the expression of several genes. Among them, genes encoding for drug transporters and metabolism enzymes are also impacted.

More precisely, NRs present a DNA binding domain (DBD), a ligand binding domain (LBD), and two activation function domains. These NRs initially stay in the cytosol and bind to corepressors, resulting in no regulatory activity. During the inflammatory signaling cascade, the corepressors are dissociated, and the ligand-NR complex is translocated into the nucleus and heterodimerizes with another NR such as PXR or it forms homodimers. Then, the recruitment of coregulators (e.g., histone acetyltransferases or CREB-binding protein), allows NRs to control the transcription event by stimulating or repressing the transcription of target genes [24]. Interestingly, the involvement of PXR has been widely demonstrated being involved in ABC transporters expression, using PXR-knockout mice. The downregulation of several drug transporters and enzymes in LPS or cytokines-treated mice was abolished in PXR-knockout mice, suggesting its involvement in regulation of drug transporters expression [25,26].

As a result, these drug transporter variations can lead to changes in both pharmacokinetics and pharmacodynamics of numerous drugs. Over the past 20 years, inflammation has been widely reported to be associated with modifications in drug-binding plasma protein levels. Various hepatic and extrahepatic drug metabolizing enzymes have also been observed to be downregulated, increasing drug exposure. Although these metabolizing enzymes could contribute more significantly to changes in drug pharmacokinetics during inflammation, the resulting changes are cumulative with the modulation of drugs efflux transporters [27,28,29,30]. Moreover, the expression of drug transporters has been reported to be modified by inflammation. All these observations underline the inflammatory mechanisms and their impact on drug exposure require extensive investigations.

## 3. Models for Studying Drug Transporter Modulation during Inflammation

Numerous models have been developed to study the effects of inflammation on drug transporter expression/functionality. These models are based on the induction of an inflammatory stimulus in rodents or cell cultures. Thereby, the fundamental characteristic of a reliable inflammation model corresponds to its ability to respond to this stimulus. This part of the review describes the different inflammation models used for assessing ABC transporter modulation.

### 3.1. In Vitro Models

Regardless of different barrier models, the in vitro modeling of inflammation is based on the stimulation of cells by proinflammatory cytokines. These models focus on the cellular effects of cytokine exposure and are limited to a key single step in the physiological process of inflammation. Generally, cells that have been characterized for being close to the human barriers are exposed to the proinflammatory cytokines for 24–72 h, modeling an inflammatory stimulus. In most cases, the ability of cells to respond to an inflammatory stimulus is determined by evaluating the secretion of other cytokines and chemokines such as IL-8 or acute phase proteins. The intestinal model Caco-2, which has been widely characterized for the evaluation of drug absorption, is the most used intestinal cell line for reproducing an in vitro inflammatory response. The exposure of Caco-2 cells for 24 h to TNF-α or IL-1β has shown an increase of IL-8 secretion, suggesting the ability of the model to establish an appropriate response [31,32]. Recent research using a 3D human model based on the culture of primary intestinal cells termed EpiIntestinal^TM^, have shown the release of IL-6 and IL-8 in response to the exposure of IL-1β, TNF-α, and IFN-γ for 24 h [33,34]. For the hepatic barrier, several models are found in the literature including the human hepatoma cell lines, HepG2 and Huh7. These models have shown the production of acute phase proteins such as serum amyloid A protein and CRP in response to the exposure of TNF-α, IL-1β, or IL-6 [35]. However, it is important to keep in mind that cells derived from a tumoral tissue may induce different expression profiles of ABC transporters, as reported in recent studies for these two models [36,37]. Another highly differentiated hepatoma cell line, HepaRG, has been characterized to be close to human primary hepatocytes. The treatment of differentiated HepaRG cells with proinflammatory cytokines induced the expression of CRP and IL-8, suggesting their sensibility to inflammatory conditions [38]. Finally, the culture of primary human hepatocytes (PHH), considered as the gold standard, has also been used for assessing drug transporter expression during inflammation. These cells are obtained from the excised hepatic tissues and isolated by enzymatic dissociation. The culturing of cells could then be performed with matrigel in a sandwich configuration or on collagen-coated plastic dishes. These human primary hepatocytes appear to be sensitive to proinflammatory cytokine exposure. Cell treatment with TNF-α, IL-1β, and IL-6 resulted also in the induction of acute phase proteins [39].

Unlike for the liver, in vitro models for assessing drug transporter expression during inflammation have not been described for the kidney. Despite the existence of several human renal cell lines such as HK-2, Caki or RPTEC/TERT1, to our knowledge none of them have been studied in inflammatory conditions. Only the culture of rat proximal tubular cells isolated from renal biopsies was used to study the modulation of drug transporters. Care needs to be taken regarding rat primary cells, which have shown different profiles of ABC transporter expression with higher expression of P-gp and a lower level of BCRP compared to human cells [40]. In the same way, many other factors can contribute to the difference in the expression of transporters within cell models, including culture conditions, number of passages or cross-laboratory variations. Thereby, it is important to have internal controls to achieve a reliable data interpretation [41,42,43].

Concerning the BBB, some models were described for in vitro studies. The brain microvascular endothelial cell line hCMEC/D3, derived from human temporal lobe microvessels and isolated from tissue excised during surgery, was used several times. These immortalized cells that closely mimic the in vivo phenotype of the BBB were exposed to proinflammatory cytokines and showed an alteration of the expression of several proteins in response to inflammatory stimuli [44,45]. Although the use of human cells is preferred, animal cells were also found in the literature. Short term exposure of isolated rat brain capillaries to proinflammatory cytokines impacted protein synthesis in response to stimuli [46,47]. In the same way, another BBB model consisting of primary porcine brain capillary endothelial cells (PBECs) was also employed [48,49]. More recently, the use of immortalized mouse brain endothelial cells, bEnd.3, in coculture contact with C6 rat astrocytes was found to assess the modulation of drug transporters during cytokine exposure [50]. From the cell culture on inserts, endothelial cells were seeded on the apical side, whereas astrocytes were seeded on the basal side of membrane inserts. This mode of culture allowed an optimal differentiation and polarization of cells that represents a key element to reproduce a BBB barrier.

### 3.2. In Vivo Animal Models

Although it is often difficult to extrapolate animal data to humans due to the potential of interspecies differences, rodent models remain a good alternative to study a biological phenomenon as a whole [51]. Up to now, animal models are the most widely used for preclinical studies to assess the impact of inflammation on drug transporters. In most cases, these models are based on the induction of an inflammatory response in rodents. The expression and/or the activity of transporters is then assessed using various techniques.

Investigations of the APR in rats or mice commonly employ the well characterized endotoxin lipopolysaccharide (LPS) [13,52,53]. Intraperitoneal or intravenous administration of endotoxin, a major cell wall component of gram-negative bacteria, induces a systemic inflammatory response characterized by symptoms such as fever, hypotension and tachycardia [52,53,54]. Generally, administration of endotoxin results in an elevation of TNF-α, IL-1β, IL-6, and IFN-γ circulation in plasma [55,56]. Another in vivo model that is also commonly employed consists in the administration of turpentine, an organic solvent distilled from pine resin. A subcutaneous or intramuscular injection of turpentine in animal models induces the formation of a dermal abscess that subsequently triggers a systemic APR [57,58,59]. As for the viral infection models, the administration of a viral-like double-stranded RNA—polyinosinic: polycytidylic acid (poly I: C) in rodents, is the most commonly approach used. This model has been especially associated with major induction of IFN. The induction of IL-6 and TNF-α was also found post poly I:C administration, leading to an APR [60]. Another way to study the impact of a viral infection found in the literature, consists in the exposition of cells to a human immunodeficiency virus-1 (HIV-1) viral envelope glycoprotein (gp120). Thus, cells were shown to undergo an inflammatory response following the stimulation [61]. Direct administration of proinflammatory cytokines in rats or mice is another method used to determine the specificity of each cytokine with respect to the modulation of transporter expression [62,63]. Many other in vivo models of inflammation are still being studied and are based on the induction of a disease involving chronic inflammatory process in rodents, such as arthritis or ulcerative colitis (UC) [64,65,66,67,68,69,70,71]. Adjuvant arthritis induced in rodents with Freund’s adjuvant has been used in several studies as an animal model of rheumatoid arthritis. This model is characterized by an inflammatory response that involves an increase in major proinflammatory cytokines and which contributes to a self-perpetuating synovial inflammation and joint destruction. For inflammatory bowel diseases, acute and chronic experimental UC models are produced in rodents by providing them water containing 5% of dextran sulfate sodium (DSS). Rodents develop acute colitis with signs of diarrhea, gross rectal bleeding, and inflammatory responses, 6–10 days after ingesting water with DSS. For the experimental chronic colitis, rodents showed a response after several consummation cycles of DSS. However, the assessment of drug transporter expression in chronic inflammatory states is still limited. The majority of research focuses on APR effects, and the characterization of chronic inflammation models still need to be developed.

### 3.3. In Vivo Human Studies to Assess Drug Pharmacokinetics Variability

In vivo human models have the main advantage to integrate all the biological processes of inflammation. However, it is difficult to identify the distinct effects of inflammation on metabolic and transport function with this type of model. Clinical studies are conducted in patients with an acute or chronic inflammatory event. In most cases, it concerns patients with a stable drug regimen in whom an inflammatory episode would have increased drug exposure. This phenomenon could result in an increased bioavailability and/or a decreased clearance. The circulating concentrations of inflammatory mediators such as CRP or cytokines (TNF-α, IL-1β, IL-6, or IL-8) have been investigated in parallel to pharmacokinetic parameters of drugs [29]. Another approach consisted in comparing pharmacokinetic parameters of drugs between patients with inflammation and healthy volunteers was also done [72]. More rarely, older clinical studies in healthy volunteers who received an inflammatory stimulus by the administration of endotoxin have also been reported for assessing drug metabolism [73,74]. With the increasing amount of data related to inflammation, more clinical studies are being conducted, especially for drugs with biological monitoring.

## 4. Inflammation-Mediated Changes on Drug Transporters

### 4.1. Intestinal Barrier

Intestinal epithelium is characterized by a heterogenous population of cells with different functions including enterocytes, Paneth cells, Tuft cells, M cells, goblet cells, and stem cells. Enterocytes represent the most abundant cells and are responsible for the function of intestinal absorption. These epithelial polarized cells delineate a basolateral pole in contact with the bloodstream and an apical pole marked by the presence of a brush border membrane in contact with the intestinal lumen. In humans, P-gp, BCRP, and MRP2 are expressed at the brush border membrane and contribute to the oral bioavailability of drugs by limiting intestinal absorption. This part summarizes all the data concerning the inflammation-mediated changes in ABC transporter expression and functionality that occur at the intestinal barrier (Table 1).

In vitro experimental studies using the human cell line Caco-2 were performed to assess the impact of inflammatory conditions on P-gp. The exposure of cells cultured on membrane porous filters to TNF-α or human recombinant IL-2 cytokines has shown a decrease of the expression of P-gp using the Western Blot technique. In the same way, the functional activity of P-gp determined by efflux of rhodamine 123 was also decreased after 48 h of TNF-α treatment. In contrast, IFN-γ exposure induced a minor but significative upregulation of P-gp mRNA levels (15%) in Caco-2 cells 24 h after treatment. This observation suggested a different pattern of modulation for each cytokine [80,81]. Interestingly, no change was observed in the integrity of an in vitro barrier. The transepithelial electrical resistance and paracellular permeabilities were not modified by cytokine exposure [80]. 

Although few studies have been performed in rodents concerning the intestinal barrier, in vitro data are in accordance with the modifications observed in vivo. LPS administration in rats induced a decrease of both P-gp and MRP2 mRNA in the ileum and jejunum 8 h after treatment using the RT-PCR technique. This regulation was associated with an increase of Il-6 and IL-1β mRNA in the same gut areas supporting the role of cytokines in drug transporter modulation [75,76,77]. Kalitsky-Szirtes et al. (2004) also showed a 50% reduced P-gp-efflux (basal to apical transport) of an antiarrhythmic drug, amiodarone, from treated rat cells. The functionality of MRP2 was also reduced by 31% in this study, as shown by a decrease in the efflux of its substrate.

For chronic conditions, arthritic rat models reported divergences about drug transporter expression during inflammation. A significant decrease of P-gp and MRP2 mRNA levels was found 21 days after the administration of an adjuvant [78]. With the same model, different consequences were observed. A slight increase in P-gp protein expression using Western Blot technique was observed 7 days after the induction of inflammation [78]. On the other hand, in vivo studies in mice with cytokine administration showed a systematic downregulation of P-gp expression and functionality [79]. Veau et al. (2002) demonstrated a decreased basal transport of rhodamine 123 across intestine in IL-2-pretreated mice with an original approach based on the use of everted gut sacs [79].

Several clinical studies with UC and Crohn’s disease patients have shown a major decrease in P-gp expression (85–89%) compared to healthy humans biopsies [83,84,85,86,87,91,92]. However, a recent study that used a targeted quantitative proteomic analysis to compare the expression of 15 transporters in healthy and inflamed humans biopsies showed no change in P-gp expression [90]. Even though a reduction in *MDR1* levels (fold change: −2.94) was observed in patients with UC compared to the healthy biopsies, no significant change in relative abundance of protein was observed [90]. It is also important to note that few studies have investigated the impact on BCRP and MRP2 expression in intestinal tissue. An analysis in UC patient biopsies revealed that the BCRP protein expression was strongly reduced by 89% compared to the control group using Western Blot and immunohistochemistry techniques [83]. It is evident that additional studies are required in order to further investigate the modulation of intestinal BCRP and MRP2 expression during inflammation.

### 4.2. Hepatic Efflux Transporters

The metabolic conversion and excretion of many drugs are performed by the most abundant cells in the liver, the hepatocytes. These polarized epithelial cells are characterized by a basolateral membrane (sinusoidal) in contact with the bloodstream and an apical membrane (canalicular) in contact with the bile canaliculus. The canalicular membrane corresponds to the excretory pole of hepatocytes and involves the presence of efflux transporters for the excretion of drugs into the bile. Among them, P-gp, BCRP and MRP2 represent the predominant canalicular drug efflux system of hepatocytes and they contribute to the hepatobiliary clearance of various classes of therapeutic compounds.

Numerous data about hepatic drug transporters regulation during inflammatory conditions have been obtained using in vitro models (Table 2). PHH and cultured human hepatoma cells, including HepG2, Huh7, and HepaRG cells, have shown a significant downregulation of P-gp, BCRP, and MRP2 mRNA levels after an exposure for 24 h to IL-6, IL-1β and IFN-γ [93,94]. The exposure of Huh7 cells to IL-6 has also been reported to reduce P-gp-mediated efflux of rhodamine 123 in accordance with the downregulation of its mRNA level [93]. Interestingly, TNF-α exposure demonstrated different patterns of regulation. This cytokine induced a decreased expression of P-gp and MRP2 in hepatoma cells HepaRG and Huh7; whereas no difference was observed in TNF-α-exposed PHH [94,95]. This observation suggests that ABC transporters are regulated differently by TNF-α in hepatoma and normal cells. Moreover, a difference between animal and human cells may also be considered. The in vitro IFN-γ exposure of primary rat hepatocytes showed a more important downregulation of P-gp expression in the range of 35–70% versus 40% in human cells [63]. These differing data underline the lack of standardization in experimental conditions. Nevertheless, in vitro assays using RT-qPCR and Western Blot analysis have shown a negative regulation of ABC transporters in all models of hepatocytes, in particular P-gp, which has been more studied.

An alteration in the expression of hepatic transporters was also demonstrated in vivo with rodent models (Table 2). The significant downregulation of hepatic P-gp expression was reported in both mice and rats as a consequence of inflammation induction. Chronic inflammation using arthritis-induced rats has also shown a significant decrease in P-gp, BCRP, and MRP2 hepatic expression in membrane fractions by Western Blot technique [70,78]. In vivo data from studies in rats seem to be in accordance with this trend, as administration of poly I:C, significantly decreased protein levels of hepatic P-gp, BCRP and MRP2 of 49, 58, and 59%, respectively [110]. In addition, the consistent data found in both in vitro and in vivo models support the link between the exposure of hepatocytes to proinflammatory cytokines and the modulation of transporter expression.

### 4.3. Blood-Brain Barrier

Brain capillary endothelial cells without fenestrations form the BBB, a highly selective permeability membrane between the blood and the brain in association with astrocytes, pericytes, and neurons [111]. Besides tight junctions that prevent the passive diffusion of small hydrophilic compounds, endothelial cells express several drug membrane transporters that limit the amount of substance penetrating the brain. The expression of ABC transporters at the apical membrane of the brain endothelium is an important element of the BBB which builds up a selective and active transport barrier. Among them, P-gp, BCRP and MRP2 have been identified to limit drug penetration into the brain.

In vitro studies using different models of BBB have provided different results on the expression of P-gp and BCRP after an exposure of cells to cytokines (Table 3). A downregulation of P-gp activity was observed without a change in protein expression after a short exposure of isolated rat brain capillaries to TNF-α (1 h); whereas, increased P-gp expression and activity were found following a more prolonged exposure (6 h) [46,47]. These results suggested a rapid decrease of functionality that did not depend on protein expression; a result that also showed a sequential modulation of P-gp at the BBB. Indeed, the time of cytokine exposure appears to be crucial for data interpretation. The expression of P-gp was downregulated by an exposure of 24 h to TNF-α of PBECs [48,50]. However, TNF-α exposure for 6 h in the same model (PBECs) showed an opposite modulation with an increased expression of P-gp. In the same way, the exposure of human cell line hCMEC/D3 to TNF-α for 72 h showed an upregulation of P-gp expression [45]. Interestingly, primary rat astrocytes treated with gp120 for 24 h, which mimics an HIV-infection, demonstrated a decreased protein expression of P-gp by Western-Blot technique [61]. These contrasting observations concerning P-gp may be partly due to the diversity of inflammatory stimuli used. For example, it is known that HIV-infection models lead to a major induction of IFN secretion compared to the other cytokines [60]. In the same way, in addition to the cytokine-dependent effect, the impact of an exposure to multiple cytokines in combination, is an important issue.

Concerning the expression and functionality of BCRP, both porcine and human models showed a downregulation in response to IL-1β exposure [45,48]. It is clear that the lack of harmonization regarding to the mode of in vitro cytokine exposure does not allow to finely interpret and compare the results. The monitoring of cytokine exposure over several hours could help to delineate divergences seen across the different inflammation models.

Interestingly, in vivo studies conducted in LPS-treated mice have shown a downregulation of P-gp function that was associated with an increase in protein expression, a result that shows the potential existence of post-translational mechanisms which require further investigation [112,113]. Moreover, increased brain accumulation of doxorubicin correlated with a decrease in P-gp protein expression in the brain of endotoxin-treated mice [114], as observed in LPS-treated rats that also showed an increase of H^3^-digoxin accumulation in the brain supporting the downregulation of drug P-gp transport at the BBB during inflammation [96]. Finally, an observational human pharmacokinetic study on patients with acute inflammatory brain injury showed a correlation between an increase in IL-6 brain exposure with the accumulation of morphine metabolites, suggesting an altered P-gp-mediated transport across the human BBB [115]. However, this clinical study presented some limitations, with most patients receiving several medications including P-gp inhibitors, such as amiodarone, that could be the source of an altered P-gp transport by drug interactions.

All these findings support that inflammation could alter drug transport across the BBB by repressing ABC transporters. Most of the studies have focused on P-gp, and the rest of predominant ABC transporters at the BBB are still poorly investigated. Faced with the clinical relevance of these phenomena, more preclinical and clinical data about BCRP and MRP2 are required.

### 4.4. Renal Efflux Transporters

Active tubular secretion, which occurs in the proximal tubule, plays a key role in the excretion of many drugs and metabolites. Polarized epithelial cells of the proximal tubule that form a tight renal barrier express several ABC transporters on the apical membrane. Many of them have been identified such as P-gp, BCRP,MRP2/4, and MATE1/2-K. They transport the substrates from cells into the lumen to be excreted via the urine. This part of the review discusses the impact of inflammation on drug renal transporters (Table 4).

Despite its clinical importance, only few studies have generated data regarding active tubular secretion under inflammatory conditions. Renal expression of transporters appears to be upregulated in vitro. The exposure of proximal tubule cells isolated from rats to TNF-α alone or to a combination of TNF-α and LPS for 24 h increased P-gp expression [116]. This result has not been confirmed in human cells or any other in vitro model. The first question that arises is whether such transporters are also similarly regulated in vivo. In vivo studies in LPS-treated mice and rats have found an upregulation of renal P-gp after 6 and 24 h of treatment [117,118]. Similarly, protein levels of renal P-gp and MRP2 but not BCRP were significantly increased in arthritis-induce rat models [70,119]. Few pharmacokinetic studies performed in rats with induced inflammation have been in accordance with this observation. Drug renal clearance could be increased during inflammation, whereas hepatic clearance was downregulated [117,120]. This phenomenon suggests the presence of a compensatory mechanism. Excretion functions would be upregulated to reduce injury caused by the accumulation of drugs [121,122]. On the other hand, some studies have shown the opposite effect with a significant decrease in P-gp expression and renal excretion in rats [123,124]. The conflicting results should be further investigated to better predict the in vivo human situation. Since most of the data obtained for renal excretion are from animal models, it would be interesting to perform drug transport studies using human renal cell models exposed to cytokines.

## 5. Impact of Inflammation on Drug Pharmacokinetics

### 5.1. Animal Data

Numerous reports on treated rodents have shown significant alterations in pharmacokinetics of drugs (Table 5). In most cases, these approaches have been based on the administration of drugs, in which pharmacokinetic parameters were monitored during inflammation. In turpentine-treated rats, the plasma concentrations of propranolol were increased after an oral administration (a 20-fold higher AUC) compared to the untreated rats [125]. Numerous other drugs have shown important modifications in pharmacokinetics during inflammation including digoxin, ciclosporin A, fexofenadine, acebutolol, doxorubicin, and verapamil [79,117,125,126]. In the same way, investigations of ^3^H-digoxin disposition are another approach commonly used to assess both the in vivo-P-gp function and drug hepatic/renal clearance. Indeed, many studies have shown a significant decrease of ^3^H-digoxin biliary clearance in rodents [101,115,116]. Taken together, preclinical data showed an increase in drug exposure during inflammation.

### 5.2. Human Data

Reports on rodents are in accordance with clinical studies in humans that also showed drug pharmacokinetic variability during inflammation (Table 6). Several classes of therapeutics are impacted including psychotropic, antiepileptic, antifungal, anticancer, and antiarrhythmics drugs [130,134,135,136]. In most studies, pharmacokinetics parameters were compared between healthy volunteers and patients with inflammatory events, but in some cases inflammatory markers such as CRP or proinflammatory cytokines were measured to establish a link between inflammation and pharmacokinetic variability. Some clinical studies have shown a correlation between high plasma CRP concentration and pharmacokinetic variability of voriconazole [137,138]. In addition, other studies establishing a link between HIV-infected patients and an induced-chronic inflammation, have shown pharmacokinetics alterations of lopinavir and darunavir. An increased exposure was observed for both these antiretroviral drugs in HIV-infected patients that was linked to the inflammatory statue [139,140]. Thereby, Seifert et al. highlighted that drug pharmacokinetic alterations during HIV-infection may be influenced by chronic inflammation associated with the disease. The dysregulated cytokine production by macrophages during HIV-infection is likely to downregulate drug metabolizing enzymes and transporters [141].

It is also interesting to note that the majority of data from humans have been provided by retrospective studies or by case reports as reviewed by Stanke-Labesque et al. (2020) [134]. In addition, several studies from inflammatory models have described a change in many steps of drug pharmacokinetics including absorption, distribution, metabolism, and elimination. These pharmacokinetic variabilities are particularly dangerous for drugs with a narrow therapeutic index as shown in Figure 1. However, the mechanistic understanding of the impact of inflammation requires a knowledge of modulation of main pharmacokinetic determinants including transporters and metabolic enzymes.

## 6. Discussion

The inflammatory response induces changes in the expression and functionality of several ABC transporters within the different physiological barriers. This modulation can impact drug disposition as shown in Figure 1. Despite numerous results available concerning the downregulation of intestinal P-gp expression, there is a lack of data about BCRP and MRP2 modulations. Conversely, the impact of inflammation on ABC transporters in hepatic cells has been well established for several years. The expression of P-gp, BCRP, and MRP2 appears to be downregulated in all in vitro and in vivo hepatic models. This conclusion is fully supported by the observation of patients suffering from liver disease involving an inflammatory event and who showed a reduced hepatic expression of ABC transporters [10,146]. Unlike intestinal and hepatic transporters, renal drug efflux transporters appear to be upregulated in most of the inflammation models. This observation suggests the existence of compensatory mechanisms between hepatic and renal drug clearance. Renal excretion of drugs may be increased in response to the decrease in biliary clearance. Brandoni et al. (2003) established that protein expression of MRP2 and organic anions transporters are increased in proximal tubule cells from rats with extrahepatic cholestasis. Thereby, this pattern occurring in renal cells may be a compensatory mechanism to increase the excretion of drugs that could not be excreted by the liver in a pathological state [121,147]. However, to date, no data are available in human renal models, which makes it difficult to predict the human in vivo situation. As for the BBB, the important divergence of the observed data leads to a difficult interpretation. These differences in the ABC transporter modulation may be explained by the multitude of models used and the conditions of cytokine exposure. Nevertheless, the decrease in P-gp and BCRP activity using probe substrates was reported in several studies, leading to a potential increase in drug brain retention during inflammation [96,112,114,148].

The first limitation in data interpretation is the lack of reports on the functional expression of transporters for all the physiological barriers. In most cases, the changes in transporter expression are observed at the mRNA level and are not always confirmed at the protein expression level. Moreover, a perfect correlation between the level of mRNA and protein expression is not always demonstrated [149,150,151]. The emerging technology of mass spectrometry-based quantitative proteomics provides a powerful tool to assess quantitative differences in protein profiles of different samples [152]. The relative quantification of ABC transporters in healthy versus inflamed tissues may be interesting to assess the impact of an inflammatory event on their protein expression. In general, mass spectrometry analysis allows the more specific quantification of proteins in various tissues [153]. Although, the mainstay of protein quantification has been the Western Blot, this technique presents some limitations, mainly related to the performance of the antibodies. Thus, it would be interesting to compare data from Western Blot and RT-qPCR analysis with data from mass spectrometry in order to increase the level of evidence regarding the modulation of ABC transporter expression.

The functionality of transporters also represents crucial information which is not clearly elucidated and that represents the final step in the investigations on drug transporter modulations. Some reports on treated rodents have shown important modifications in the pharmacokinetics of several drugs during inflammation including digoxin, ciclosporin A, fexofenadine, acebutolol, doxorubicin, verapamil, and propranolol [117,125,126,131]. Nevertheless, further studies are required to better identify pharmacokinetic changes due to the modulation of ABC transporters. It is important to note that variability in drug exposure during inflammation is the result of modulation of the expression and activity of several pharmacokinetic players. Cytochrome enzymes have a major role in this variability. Although it is difficult to establish the proportion of modulation related to transporters, many drugs have a high affinity for these transporters. In addition, it is known that CYP3A and P-gp are both regulated by the same nuclear receptor, PXR [154,155]. Thus, it would be interesting to integrate all the pharmacokinetic actors in physiologically-based pharmacokinetic modeling (PBPK) to establish more complete in vitro–in vivo correlations, as suggested by Simon et al. (2020) [156]. Conducting prospective and retrospective studies in humans could provide more evidence, by establishing for example, a link between the level of inflammatory markers such as CRP and a modulation of drug exposure [130,138].

Another major limitation in the interpretation of data is the lack of human models for studying drug transporter modulations. It is clear here that most results have been provided by rodent models. However, it is known that the expression and functionality of ABC transporters differs between humans and rodents [157,158]. Human in vitro models could therefore represent suitable and relevant tools in the evaluation of drug transport modulations during inflammation. These models consist of exposing cells to proinflammatory cytokines and present the main advantage to specifically study the effect of inflammation on transporters. The first question that arises is whether cells are also similarly in vivo regulated by these cytokines in humans. Among all the in vitro models used for assessing drug transporter modulations, few have been pharmacologically characterized. Although several modulations were observed in vitro for the expression and functionality of drug transporters, the lack of standardization is an important issue for data interpretation. The in vitro effects of cytokine exposure consist of the activation of signaling cascades, the increase of transcriptional activity and IL-8 secretion. However, the literature shows a lack of homogeneity in experimental conditions. The stimulus concentration, exposure duration and nature of cells are not standardized. Regarding the concentration of cytokines used, among all in vitro studies cited, the concentrations are in the range of 10 to 100 ng/mL. Although these concentrations are nearly 1000 times higher than those found in humans (10–100 pg/mL) during an inflammatory event, it is difficult to observe an effect in vitro at low concentrations [7,159]. Nevertheless, it would be judicious to expose the cells to increasing concentrations of cytokines in order to accurately determine the half maximal effective concentration (EC50) for each cytokine. Moreover, although cytokines are individually known to induce a cellular response, data on their concerted action as occurring in vivo have still not been elucidated in vitro. Interestingly, Van De Walle et al. (2010) investigated experimental conditions to develop an in vitro model of APR during intestinal inflammation using Caco-2 cells. They have shown that the combination of IL-1β, TNF-α, and IFN-γ accurately mimics intestinal inflammatory processes compared to an individual exposure to each cytokine [31]. The nature of cells is also an important parameter for in vitro modeling of inflammation. Although, healthy primary human cultures usually reflect in vivo situations well, numerous animal or tumoral cells are used and care should be taken regarding data interpretations. Thereby, the standardization of culture and cytokine exposure conditions is a crucial requirement to allow a more reliable interpretation of data. Firstly, the development of a reliable in vitro barrier model involves the pharmacological characterization of models by assessing barrier properties and drug transporters expression. Secondly, the in vitro exposure to cytokines needs to be standardized for reproducing acute and chronic inflammatory conditions. To this end, the experimental conditions should be precisely defined. The concentration, duration and combination of cytokines exposure should be characterized. Obviously, in vitro acute stimulation requires different conditions compared to the chronic response. Cell exposure to cytokines should not exceed 72 h to accurately mimic an APR. Hackett et al., who focused on cytokines kinetics after LPS-stimulation of human lung tissues, showed that the maximum cytokines levels were reached 24 h and 48 h after stimulation for TNF-α and IL-6, respectively [160]. In the same way, Hackett et al., also showed that the anti-inflammatory cytokine IL-10 had opposite kinetics with a progressive increase in concentrations from 48 h post-stimulation, suggesting a return to a non-inflammatory state [160,161]. In addition, the exposure of cells to LPS or direct contact of cells with pathogenic agents appear to be a good alternative for inducing an APR. Conversely, conditions of chronic inflammation in vitro require a continuous stimulation of cells to proinflammatory cytokines. For example, Seidelin et al., have performed a repeated exposure of intestinal cells (HT29) to TNF-α and IFN-γ for 9 weeks to mimic chronic intestinal mucosal inflammation [162]. However, most of the studies focusing on the modulation of transporters during chronic inflammation have used animal models. The use of in vitro approaches to study the impact of chronic inflammation would be interesting to study the modulation of transporters and should be further used. Nevertheless, this type of approach requires experimental validation of chronic exposure by evaluating cellular markers of stimulation, such as IL-8 secretion, during the entire exposure period.

Lastly, care should be taken regarding the techniques to assess modulations of functional expression of ABC transporters. Reference techniques to quantify a variation in the expression should be coupled with mass spectrometry analysis. In addition, functionality studies, especially on the choice of substrate molecules, should also be clearly defined. Fluorescent tracer dyes including rhodamine 123 and calcein-AM represent an important class of subcellular probes and allow the examination of membrane transport by P-gp. However, it is known that both protocol assays and cell lines may influence the interpretation of transport assays [163,164,165]. The influence of these parameters should be more studied and a comparison with a drug substrate reference such as digoxin should also be assessed.

## 7. Conclusions

The inflammatory mechanisms found in a vast number of acute and chronic diseases induce changes in the expression and functionality of many ABC transporters. These transporters play an important role in the disposition of several drugs. Inflammation could therefore be an important determinant of interindividual variability in drug pharmacokinetics. The release of proinflammatory cytokines have been widely associated in the regulation of ABC transporters. Preclinical data suggest that the expression of P-gp, BCRP, and MRP2 is downregulated in most physiological barriers, while it appears to be upregulated in the kidney. Some divergent results were observed between the different models used, this is partly due to the lack of standardization in the development of preclinical models. The choice of several parameters needs to be investigated including cytokine exposure conditions and the nature of cells. Moreover, additional techniques such as mass spectrometry may give more accurate data on quantitative inflammation-mediated changes in ABC transporters. For the future, to complete in vitro and in vivo rodent data, prospective and retrospective studies in human subjects should be performed to add further evidence to drug pharmacokinetic variability during inflammation.

## Figures and Tables

**Figure 1 pharmaceutics-13-01544-f001:**
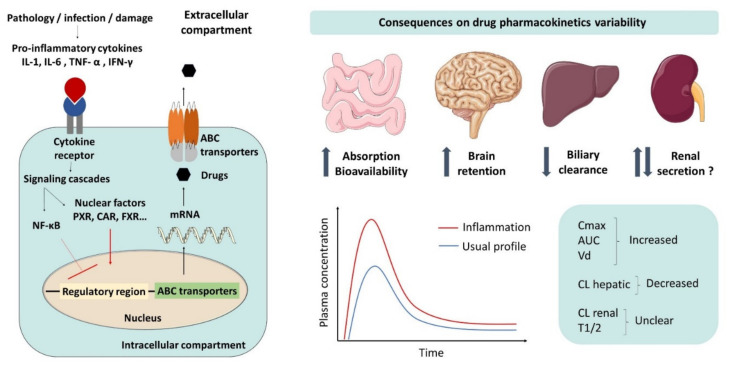
Modulation of ABC transporters during inflammation by proinflammatory cytokines and consequences on drug pharmacokinetic variability. Pathology, infection, or tissue damage stimulate the release of proinflammatory cytokines which interact with their receptors on epithelial or endothelial cell membranes, leading to the activation of signaling cascades and to the modulation of ABC transporter expression. These molecular pathways result in the activation of NF-κB, involving the inhibition of the heterodimerization of the nuclear receptor retinoid x receptor (RXR) with other nuclear factors such as pregnane X receptor (PXR), constitutive androstane receptor (CAR), and farnesoid X receptor (FRX). Consequently, up- or downregulation of ABC transporters may affect drug exposure by increasing intestinal absorption and brain retention and by decreasing drug clearance.

**Table 1 pharmaceutics-13-01544-t001:** Summary of the inflammation-mediated changes in transporter expression and function in intestine.

Transporter	Model	Experiment	Technique	Gut Area	Gene Expression	Protein	Function	References
P-gp	LPS-treated mice and rats	Endotoxin administration	RT-PCR Western Blot	Jejunum Ileum	↓ mRNA (30–78%)	↓ (20–78%)	↓ (41–56%)	[75,76,77]
			Immunostaining		ND	Decrease staining	ND	[75]
	Arthritic rats	Freund’s adjuvant administration	RT-PCR Western Blot	Jejunum Ileum	↓ mRNA (30%)	No change	ND	[70]
					↓ mRNA (52%)	ND	ND	[78]
	Mice	Cytokines administration	RT-PCR Western Blot	Ileum Duodenum	↓ mRNA (40–57%)	↓ (48%)	↓ (30%)	[79]
	Caco-2	TNF- α exposure	RT-PCR Western Blot	-	↓ mRNA (56%)	↓ (29%)	↓ (48%)	[80]
		IFN- γ exposure			↑ mRNA (29%)	↑(22%)	No change	[80,81]
		Hr-IL2 exposure			↓ mRNA (100%)	ND	↓ (21%)	[82]
	Human	Biopsies on ulcerative colitis patients	RT-PCR	Colon/Ileum	↓ mRNA (50–89%)	ND	ND	[83,84,85,86,87]
			RT-PCR Western Blot Immunohistochemistry	Colon	↓ mRNA (50%)	Decrease staining	ND	[88,89]
			RT-PCR / Proteomic	Colon	↓ mRNA (67%)	No change	ND	[90]
		Biopsies on Crohn’s disease patients	RT-PCR	Colon	↓ mRNA (64%)	ND	ND	[87]
MRP2	LPS-treated rats	Endotoxin administration	RT-PCR	Jejunum Ileum Colon	↓ mRNA (50%)	ND	↓ (31%)	[76]
	Arthritic rats	Freund’s adjuvant administration			↓ mRNA (29%)		ND	[78]
BCRP	Arthritic rats	Freund’s adjuvant administration	RT-PCR	-	No change	ND	ND	[78]
	Human	Biopsies on ulcerative colitis patients	RT-PCR	-	↓ mRNA (30–89%)	ND	ND	[83,85,90]

↓↑ significant repression/upregulation compared with control. ND, not determined; Hr-IL2, Human recombinant-IL2.

**Table 2 pharmaceutics-13-01544-t002:** Summary of inflammation-mediated changes in transporter expression and function in liver.

Transporter	Models	Experiment	Technique	Gene Expression	Protein	Function	References
P-gp	LPS-treated mice and rats	Endotoxin administration	RT-PCR Western Blot	↓ mRNA (55–98%)	↓ (40–80%)	↓ (50–75%)	[96,97,98,99,100,101,102,103]
	Arthritic rats	Freund’s adjuvant administration	RT-PCR Western Blot	↓ mRNA (50–52%)	↓ (20%)	↓ (70%)	[70,78]
			Immunostaining	ND	Decreased staining	↓ (37%)	[104]
	HuH7 cells	IL-6 exposure	RT-PCR Western Blot	↓ mRNA (40%)	↓ (20%)	↓ (20%)	[93]
	HepaRG cells	TNF-α exposure	RT-PCR	↓ mRNA (55%)	ND	ND	[94]
		IL-6 exposure		↓ mRNA (52%)			
	Primary Human Hepatocytes	IL-6 exposure	RT-PCR Western Blot	↓ mRNA (37%)	No change	ND	[95,105]
		IL-1β exposure		No change			[95]
		TNF-α exposure		↓ mRNA (40%)			
	Rat hepatocytes	IL-1β exposure	RT-PCR Western Blot	ND	↓ (40%)	↓ (34%)	[63]
		IL-6 exposure		↓ mRNA (60%)	↓ (55%)	↓ (36%)	
MRP2	LPS-treated mice and rats	Endotoxin administration	RT-PCR Western	↓ mRNA (55–93%)	↓ (58–70%)	ND	[101,102,106]
	Treated mice and rats	Turpentine injection	RT-PCR Northern Blot	↓ mRNA (30–40%)	ND	ND	[107]
	Arthritic rats	Freund’s adjuvant administration	RT-PCR Western Blot	↓ mRNA (29%)	↓ (50%)	ND	[70,78]
	HepG2 cells	IL-1β exposure	RT-PCR	↓ mRNA (40%)	ND	ND	[108]
		TNF-α exposure		↓ mRNA (40%)			
BCRP	Primary Human Hepatocytes	IFN-γ exposure	RT-PCR	↓ mRNA (41%)	ND	ND	[109]
	HepaRG cells	IL-6 exposure		↓ mRNA (50%)	ND	ND	[94]

↓, significant repression compared with control. ND, not determined.

**Table 3 pharmaceutics-13-01544-t003:** Summary of inflammation-mediated changes in transporter expression and function in the blood-brain barrier.

Transporter	Models	Experiment	Technique	Gene Expression	Protein	Function	References
P-gp	LPS-treated mice	Endotoxin administration	RT-PCR Western Blot	↑ mRNA (40%)	↑ (44%)	↓ (10%)	[112,113]
	LPS-treated rats			↓ mRNA (50%)	ND	↓ (160%)	[96]
	hCMEC/D3 cells	IL-6 for 72 h	RT-PCR Western Blot	↓ mRNA (21–43%)	No change	↓ (28%)	[45]
		TNF-α for 72 h		↑ mRNA (49%)	↑ (33%)	No change	[45]
		IL-1β for 72 h		↓ mRNA (14%)	No change		
	Rat brain capillaries	TNF-α for 1 h	Western Blot	ND	↓ (50%)	↓ (50%)	[46]
		TNF-α for 6 h	Western Blot Immunofluorescence	ND	↑ (50%)	ND	[47]
	Porcine Brain endothelial cells PBECs	TNF-α for 6 h for 24 h for 48 h	RT-Qpcr Western Blot	No change	↑ (27%) ↓ (10%) ↓ (48%)	ND	[48]
		IL-1β for 24 h		No change	↓ (45%)	ND	[48]
		IL-1β for 24 h	Western Blot	ND	↑ (180%)	↑ (25%)	[49]
	Coculture system bEnd.3/astrocytes C6	IL-6 for 24 h	ELISA	ND	↓ (42,6%)	No change	[50]
		TNF-α for 24 h		ND	↓ (37%)	No change	[50]
BCRP	hCMEC/D3 cells	IL-6 for 72 h	RT-PCR Western Blot	↓ mRNA (39–45%)	No change	↓ (96%)	[34,45]
		TNF-α for 72 h		↓ mRNA (33%)		↓ (69%)	[36]
		IL-1β for 72 h		↓ mRNA (31%)		↓ (57%)	[45]
	PBCECs	TNF-α for 6 h for 24 h for 48 h	RT-PCR Western Blot	↓ mRNA (32%) No change No change	No change ↓ (27%) ↓ (42%)	No change ↓ (32%) ↓ (28%)	[48]

↓↑ significant repression/upregulation compared with control. ND not determined.

**Table 4 pharmaceutics-13-01544-t004:** Summary of inflammation-mediated changes in transporter expression and function in kidney.

Transporter	Models	Experiment	Technique	Gene Expression	Protein	Function	References
P-gp	LPS-treated rats	Endotoxin administration	RT-PCR Western Blot	↑ mRNA (28%)	↑ (39%)	ND	[116,117]
			RT-PCR	↓ mRNA (37%)	ND	ND	[118]
	Arthritic rats	Freund’s adjuvant administration	RT-PCR Western Blot	↑ mRNA (42%)	↑ (32%)	No change	[70]
	Rat proximal tubule cells	TNF-α exposure	RT-PCR Western Blot	↑ mRNA (29%)	↑ (43%)	↑ Increase	[116]
MRP2	LPS-treated rats	Endotoxin administration	RT-PCR Western Blot	↑ mRNA (22–49%)	ND	ND	[118]
	Arthritic rats	Freund’s adjuvant administration			↑ Increase	ND	[70]
BCRP	Arthritic rats	Freund’s adjuvant administration	RT-PCR Western Blot	↑ mRNA (42%)	No change	ND	[70]

↓↑ significant repression/upregulation compared with control. ND, not determined.

**Table 5 pharmaceutics-13-01544-t005:** Summary of animal data about the impact of inflammation on drug pharmacokinetics.

Drugs	Models	Pharmacokinetic Modulation	References
Propranolol	TT rats	↑ AUC (1900%)	[125]
	AA rats	↑ AUC (2900%)	[127]
Acebutolol	AA rats	↑ AUC (150%)	[100]
Ciclosporin A	DSS-induced colitis in rats	↑ AUC (55%)	[128]
Fexofenadine	LPS-treated rats	↑ AUC (40%)	[126]
Doxorubicin	LPS-treated rats	↑ AUC (27%)	[117]
Verapamil	RA patients	↑ AUC (300%)	[129]
Perampanel	Patients with inflammation event	↑ AUC (101%)	[130]
Digoxin/H3-digoxin	Cytokines-treatment in mice	↑ AUC (40–65%)	[79,131,132,133]
	LPS-treated rats	↑ AUC (159%)	

↑, significant decrease/increase; TT rats, turpentine-treated; AA rats, adjuvant-arthritis; DSS, dextran sulfate sodium; LPS, lipopolysaccharide; RA, rheumatoid arthritis.

**Table 6 pharmaceutics-13-01544-t006:** Summary of the impact of inflammation on drug pharmacokinetics in humans.

Drugs	Study Type	Patient Characteristics	Pharmacokinetic Modulation	References
Perampanel	Retrospective comparative study	Patients with inflammation event	↑ AUC (101%)	[130]
Clozapine	Retrospective comparative study	Patients with inflammation event (CRP ≥ 5 mg/L)	↑ exposure (48%)	[135]
Quetiapine		Patients with inflammation event (CRP ≥ 5 mg/L)	↑ exposure (11.9%)	
Risperidone		Patients with inflammation event (CRP ≥ 5 mg/L)	↑ exposure (58.4%)	
Risperidone	Case report	Case reported Patient with inflammation event (CRP ≥ 120 mg/L)	↑ exposure (263%)	[136]
Risperidone	Case report	Case reported Patient with inflammation event (CRP ≥ 110 mg/L)	↑ exposure (161%)	
Lopinavir	Prospective comparative study	HIV-infected patients with retroviral treatment or not	↑ AUC (14%)	[141]
Darunavir	Prospective comparative study	Healthy versus HIV-infected patients	↑ AUC (25–30%)	[142,143,144,145]

↑, significant increase.

## Data Availability

Not applicable.

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
