# Peer review of "Inflammation Induces Changes in the Functional Expression of P-gp, BCRP, and MRP2: An Overview of Different Models and Consequences for Drug Disposition"

_pharmaceutics, 2021, doi:10.3390/pharmaceutics13101544_

Round 1

Reviewer 1 Report

This is an important and well written review about the effects of inflammation on ABC transporter expression. There are some points which should be considered:

  1. The first page includes instructions to authors - should be removed
  2. The table of contents is falling apart, the tables are somehow inserted between the test and discussion
  3. On page 3, gene names should be italics
  4. in Tables 1 and to - decrease stainig should be changed to decreased staining
  5. Conclusions - Page 21 - vast number of diseases
  6. choice of several...needs..
  7.  

Author Response

Thank you for your feedback.

Please find below answers :

  1. Instructions to authors have been removed on the first page
  2. The table of contents has been restructured
  3. On page 3, genes ABCB1, ABCG2 and ABCC2 are now in intalic.
  4. "Decrease" has been replaced by "Decreased" un Table 1 and 2. 
  5. Conclusions - Page 21 - Line 91 disease" has been replaced by "diseases". 
  6. Conclusions - Page 21 - Line 100 "need" has been replaced by "needs"

Kind regards. 

Reviewer 2 Report

The authors of this manuscript provided a comprehensive review on the effects of proinflammatory cytokines on the expression and activity of several ABC transporters critical for drug disposition, both in vitro and in vivo. This is a timely and informative review on an important topic. A few comments need to be addressed to enhance this review article.

  1. Sections 2 and 3 were all assigned to 1. Please correct.
  2. Sub-section 2.2. Please cite references that describe how cytokines regulate gene expression (drug transporters or drug metabolizing enzymes) through nuclear receptors, not just references for signaling cascades and nuclear receptors in general. Also, provide a concise description in this section of the current understanding of how cytokines regulate the expression of target genes through nuclear receptors such as PXR, CAR.
  3. Section 2. In addition to chronic diseases, the authors should include infectious diseases such as vial (i.e. HIV) and bacterial infections that also produce cytokines. Accordingly, the authors should add examples relevant to infectious diseases in ALL subsequent sections. This is a very important piece of information the authors missed, and there have been quite a lot of studies out there in the literature.
  4. Section 4. In all of the in vitro cell-based studies, can the authors provide information regarding cytokine concentrations used? In another word, were the cytokine concentrations used physiologically relevant? Was there a concentration-dependent effect? How about the effect of a combination of various cytokines as in vivo in humans? Were cytokine concentrations changing (i.e. declining due to metabolism) during the period of incubation with cells?
  5. This reviewer suggests that the authors add a Table 6 for human PK data. Table 5 can be only for animal PK. There seems to be quite a lot of human data as described in section 5.2. All of these human data may be summarized in Table 6.
  6. Title of tables should be placed on top of each table.
  7. Can the authors provide some specific details as to what in vitro conditions would reflect acute and chronic inflammation separately?
  8. At the end of Discussion, the authors imply that inflammatory mediators such as cytokines could be used as reliable biomarkers for drug disposition. This reviewer suggests this sentence be deleted. Such a statement may not be made unless there is clear evidence to say that.    

Author Response

Kind regards.

Reviewer 3 Report

This is a well-researched review article and I found it easy to read and thorough.

That being said, there are a lot of edits to be made.  I have highlighted these in the text of the paper and made corresponding comments.  Please check these and edit the paper accordingly.

The major changes that are required are the conclusion/discussion statements made throughout the document.  For example, the differences in results between different publications are attributed to 'lack of standardization'.  How can one do this?  Different models are used depending upon the lab, the goal of the study, the available literature at that time etc.  There is no attempt made to figure out the differences and what could be causing the variability.  Or what standardization would look like.  The authors should either provide solution or stay away from such statements.

Section 3.3. starts with ‘In vivo human models have the main advantage to integrate all the biological processes of inflammation.’  How do the authors plan to use humans to study transporters in models of inflammation?  Has this ever been done?  These generalized statements need to be removed.

The title should be changed, as the review is on 3 ABC transporters and the title should reflect this.

It seems that some articles referenced were not read/understood e.g. PXR and CAR are not the ‘principal actors during inflammation’ – they play a role.

The authors use mdr1 and MDR1 interchangeably – the former is for rodents and the latter for human.  Please check.

The discussion needs a major overhaul – I have provided many, many suggestions in the text.  Please check and make appropriate changes.

The very last sentence ‘Therefore, inflammatory mediators could represent reliable biomarkers to guide drug selection and dosage requirements’ is the most confusing.  It is made without any context and is not what is discussed in the paper. Which of the mediators would guide selection of drugs?   And how?

The references need to be edited – I saw many mistakes and highlighted a few.

Author Response

Kind regards.

Round 2

Reviewer 2 Report

There are a few minor typos and grammatical error as below. 

1) Section 2.2. Second paragraph. ", allows to the NRs to control..."

2) Section 5.2. Second paragraph. The reference "Stanke-Labesque et al." should be cited with a reference number.

3) Table 6. "VIH" should be "HIV"     

This reviewer suggest the authors carefully read through the manuscript to see if there are any similar typos or grammatical errors that may be missed.  

Author Response

Dear Editor-in-Chief and reviewers,

Many thanks for having provided us the opportunity to revise our draft entitled “Inflammation induces changes in the functional expression of ABC transporters : an overview of different models and consequences for drug disposition”, submitted as a review in Pharmaceutics.

You will find below a response to every comment, and the remaining modification in the revised version of the draft.

We hope that you will be happy with the revised version and will find it be acceptable for publication in its revised form.

Kind regards,

Sonia Saib and Xavier Delavenne

__________________________________________________

1) Section 2.2. Second paragraph. ", allows to the NRs to control..."

   Answer : "Allows to the NRs" has been replaced by "Allows NRs..."

2) Section 5.2. Second paragraph. The reference "Stanke-Labesque et al." should be cited with a reference number.

   Answer : reference number has been added. 

3) Table 6. "VIH" should be "HIV"  

   Answer : all typos have been replaced by the correct word. 

Reviewer 3 Report

The edits made have improved the paper and I have no more suggestions.

Author Response

Dear Editor-in-Chief and reviewers,

Many thanks for having provided us the opportunity to revise our draft entitled “Inflammation induces changes in the functional expression of ABC transporters : an overview of different models and consequences for drug disposition”, submitted as a review in Pharmaceutics.

We hope that you will be happy with the revised version and will find it be acceptable for publication in its revised form.

Kind regards,

Sonia Saib and Xavier Delavenne

______________________________________

Remark : The edits made have improved the paper and I have no more suggestions.

Answer : Thank you for your positive feedback.